# Description and Validation of a Carbon Monoxide and Nitrous Oxide Instrument for High-Altitude Airborne Science (COMA)

Emma L. Yates[1,2], Levi M. Golston[1,3], James R. Podolske[1], Laura T. Iraci[1], Kristen E. Okorn[1,2,3], Caroline Dang[1,2,3], Roy R. Johnson[1], James Eilers[1], Richard Kolyer[1], Ian Astley[4,*], J. Brian Leen[4,*]

[1] NASA Ames Research Center, Moffet Field, CA, USA
[2] Bay Area Environmental Research Institute, Moffett Field, CA, USA
[3] NASA Postdoctoral Program, NASA Ames Research Center, Moffett Field, CA, USA
[4] ABB Group, San Jose, CA, USA
[*] Now at Nikara Labs Inc., Mountain View, CA, USA

*Correspondence to*: Emma L. Yates (emma.l.yates@nasa.gov)

**Abstract.** In this work, we describe development of the Carbon monOxide Measurement from Ames (COMA) instrument for measurement of carbon monoxide (CO) and nitrous oxide ($N_2O$) aboard NASA's WB-57 high altitude research aircraft. While COMA has previously flown in the cabin of the NASA P-3 platform, here the instrument was modified to operate in a significantly different environment- an unpressurized pallet flying primarily above 12 km (40,000 ft). Modifications were made to the laser to allow for detection of CO and $N_2O$, ruggedization and thermal management were addressed, and a calibration system was designed to quantify the measurement stability in-flight. Testing was conducted in a thermal vacuum chamber to mimic anticipated ambient conditions experienced inside the WB-57 pallet bay and found electronic components remained within thermal limits. COMA successfully operated during nine unattended transit flights to and from South Korea and fifteen research flights during NASA's Asian summer monsoon Chemical & CLImate Project (ACCLIP) 2022 campaign, which was focused on studying the Asian summer monsoon anticyclone in the Western Pacific. The CO measurement has an overall uncertainty ranging between 4.1 ppb (at 50 ppb CO) and 5.6 ppb (at 200 ppb CO). $N_2O$ has an overall uncertainty of 2.7 ppb (at 320 ppb $N_2O$). In addition, COMA observations were compared with two other in-situ CO instruments co-located on the WB-57: Carbon Monoxide Laser Detector (COLD) 2 and Airborne Carbonic Oxides and Sulfide Spectrometer (ACOS). Comparisons for 15 flights during the ACCLIP campaign indicate a range in slope of 1.10–1.15 for COLD2 vs. COMA and 0.94–1.10 for ACOS vs. COMA.

## 1 Introduction

The transport of trace gases and aerosols into and within the upper troposphere and lower stratosphere (UTLS) during the Asian summer monsoon was the focus of the Asian summer monsoon Chemical & CLImate Project (ACCLIP) field

campaign in summer 2022 (Honomichl & Pan, 2020; Pan et al., 2022, Pan et al., 2025). Carbon monoxide (CO) and nitrous oxide ($N_2O$) were key measurements during ACCLIP due to their long atmospheric lifetimes. CO is used as a tracer of boundary layer air transport and to infer airmass age (e.g. Pan et al., 2016; Park et al., 2009). Additionally, $N_2O$ is a

dominant ozone-depleting substance (Ravishankara et al., 2009) with surface sources and stratospheric loss (Tian et al., 2020), which can indicate in-mixing of aged stratospheric air (e.g., Gonzalez et al., 2021; Hintsa et al., 1998). As such, understanding the transport mechanisms and behavior of CO and $N_2O$ during the Asian summer monsoon is crucial for evaluating their impact on regional and global climate.

Given its importance, CO has been measured in numerous airborne field campaigns using different aircraft platforms and

several different measurement techniques. Established airborne measurements of CO and $N_2O$ with fast instrument response time (seconds) frequently use high-precision infrared (IR) spectroscopic techniques to create spatially and temporally dense datasets. Existing spectroscopic sensor systems operating in the mid-IR use optical sources such as Quantum Cascade Lasers (QCLs) or Interband Cascade Lasers (ICLs) in conjunction with path length enhancement techniques such as multi-pass cells [Gvakharia, Vicicani, etc.] or Off-Axis Integrated Cavity Output Spectroscopy (OA-

ICOS) [Kloss] (Gvakharia et al., 2018; Kostinek et al., 2019; Pitt et al., 2016; Viciani et al., 2018). Systems operating in the near-IR and utilizing Cavity Ring-Down Spectroscopy (CRDS) have also been fielded (Filges et al., 2015). A detailed overview of these different methods can be found in Zellweger et al. (2012).

This paper describes the work undertaken to convert NASA's Carbon monOxide Measurement from Ames (COMA), a laboratory-based OA-ICOS instrument (Los Gatos Research (now ABB Ltd.)), into an airborne instrument capable of high-

altitude (18 km) airborne measurements of CO and $N_2O$. The airborne operational requirements were that COMA had to operate autonomously in an unpressurized, unheated payload bay of NASA's WB-57 aircraft while flying at altitudes up to 18 km for up to 5 hours of flight time during the ACCLIP campaign. Restrictions on the instrument design included: space/size, low operating pressure (~75 hPa), low in-flight temperature (-20 °C), high pre- and post-flight temperature (>30 °C) and humidity, and requirements for accurate and precise CO and $N_2O$ measurements.

This work details the instrument testing, modifications, and final design of COMA, an instrument designed to meet operational requirements necessary for high altitude flights on NASA's WB-57. We also describe COMA's in-flight performance and operation and briefly discuss observations of CO and $N_2O$ in the UTLS region during the Asian summer monsoon in summer 2022.

## 2 Instrument Design and Modifications

The operational requirements to fly in one half of a pallet for the payload bay of NASA's WB-57 included instrument size restrictions to physically fit within the space provided, inlet and operating pressure as low as 50 Torr (~75 hPa) when flying at 18 km, low in-flight temperature (-20 °C), but high pre- and post-flight temperature (>30 °C) and humidity during the ACCLIP field campaign. Operation in an unpressurized UTLS environment presents multiple challenges to instrument

stability and performance, since the instrument is designed for ground-based operation. To address these challenges, the COMA instrument underwent significant modifications, followed by laboratory and environmental chamber testing to simulate instrument behavior under expected flight conditions.

## 2.1 Instrument Description

COMA is based on a laboratory-oriented OA-ICOS instrument (Los Gatos Research (now ABB Ltd.) GLA251-N2OCM) which detects a CO absorption feature as well as nearby $N_2O$ and $H_2O$ absorption features. The original instrument was manufactured for NASA Ames Research Center in December 2015. The un-modified instrument flew on NASA's P3 aircraft in a 19" rack within the pressurized cabin during the ObseRvations of Aerosols above CLouds and their intEractionS (ORACLES) field campaign in 2018, providing observations of CO and $CO_2$ on 41 flights up to 7 km altitude (Redemann et al., 2021).

In preparation for ACCLIP, some initial instrument modifications were made, including a computer stack upgrade and replacing the laser to one with increased sensitivity to CO. Changing the laser allowed for $N_2O$ observations, but at the expense of removing the $CO_2$ channel. This was to accommodate the low values of CO expected to be observed in ACCLIP's UTLS-focused campaign. Other modifications included the installation of higher conductance vacuum control valves, increasing the measurement range of the sample gas temperature thermistor, and modification of the secondary laser temperature controller setpoint for the expected thermal environment. Sample cell pressure was adjusted to operate at 52.8 torr (70.4 hPa) to accommodate ambient UTLS pressures. Enclosure heaters and two 150 W box fans were added to the COMA payload to help reduce condensation on the optical windows and to stabilize the COMA enclosure temperature.

### 2.1.1 Payload Design

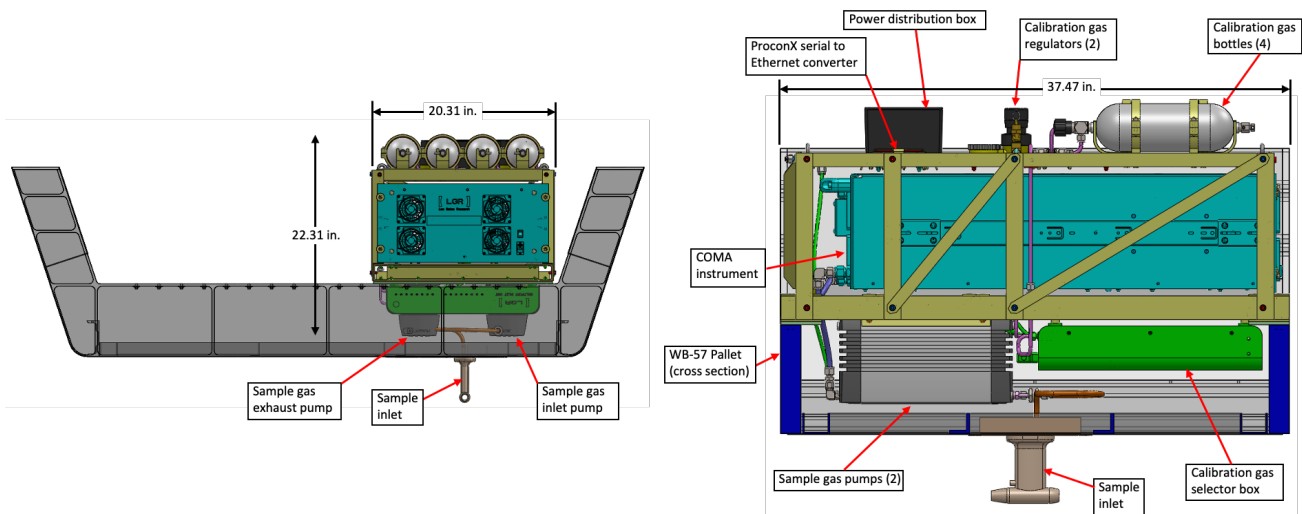

**Figure 1: COMA payload design (right) and COMA payload installed in NASA's WB-57 payload pallet (looking aft) (left). The engineering drawings show the layout of COMA within NASA's WB-57 payload bay including location of the main COMA instrument (teal) and the two sample pumps (sample gas pumps (2)) the calibration system (four calibration standards (calibration gas bottles (4)) and two regulators (calibration gas regulators (2)) and the MIU (calibration gas selector box)), power distribution box and sample inlet (bottom) and a cross-sectional view of COMA within the permitted envelope, including COMA**
**dimension details (inches) (left).**

COMA was installed in a pallet for upload into NASA's WB-57 payload bay, as shown in Figure 1. COMA's external mounting structure (chassis) was Alodine treated aluminum and featured side rails to allow for easy installation. The total assembled payload (excluding the pallet) weighed 97.8 kg. The instrument was run from 115 VAC, 60 Hz power from the aircraft. A 28 VDC line for relay activation to power the instrument from the Experimental Control Panel (ECP) was added.
The power distribution box contained breakers for the inlet heater, COMA analyzer, commercial multiport inlet unit (MIU), inlet pump, and a master switch. The external pump was controlled by COMA and was the default pump used (COMA switches from operating using an internal to external pump on initial startup). Addition of an 8-channel thermocouple temperature data logger (Madgetech TCTempX8LCD and Type K thermocouples) allowed for measurements of the thermal operating conditions experienced in-flight by COMA as shown in Figure 2.

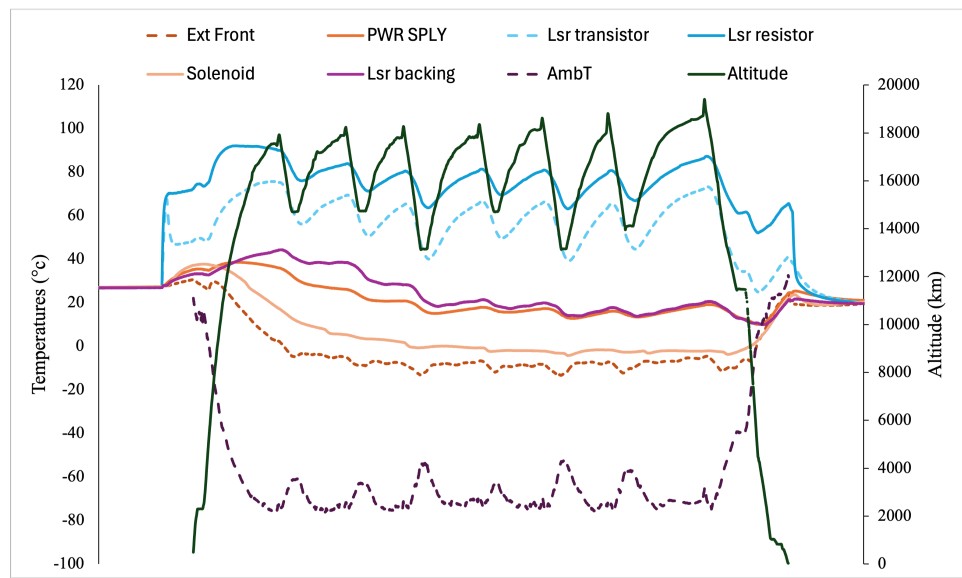

**Figure 2: Typical operating thermal conditions experienced by COMA during ACCLIP research flights, data from 21 August 2022 (Research Flight # 11). The black line shows the aircraft altitude (right axis); other lines show measured temperatures at multiple locations inside and outside the COMA chassis. Orange-red (dash) = external front (Ext front), orange = power supply (PWR SPLY), light blue (dash) = laser transistor (Lsr transistor), blue = laser resistor (Lsr resistor), light orange – solenoid,**
**purple = laser backing, purple (dash) = ambient temperature (Amb T).**

### 2.1.2 Flow System

COMA used an inlet probe, which has previously flown on NASA's WB-57, from NASA Goddard Space Flight Center known as the "CAFE Inlet" (St. Clair et al., 2019) mounted on the underside of the pallet. The inlet includes a cartridge heater (SunRod) controlled to 30 °C by a proportional controller (Minco CT335). Inside the payload bay, Teflon FEP or stainless-steel tubing carries the sample flow from the inlet to the instrument via an inlet diaphragm pump (KNF Group, NF N90 APE-W), relief valve (TAVCO 20 psia), and high-flow solenoid valve. The solenoid valve directs the sample air to one of the eight available ports on the MIU, which is programmed to control COMA's air sampling and calibration sampling pattern (discussed in Section 4.3). The inlet diaphragm pump, along with an internal (to COMA) pump and the external (to COMA) exhaust pump (KNF Group, NF N90 APE-W) pull the sample through COMA and exhaust it to the ambient atmosphere. COMA contains internal valves that maintain a pressure of 52.8 Torr within COMA's sampling cell. A diagram of COMA's flow system is shown in Figure 3.

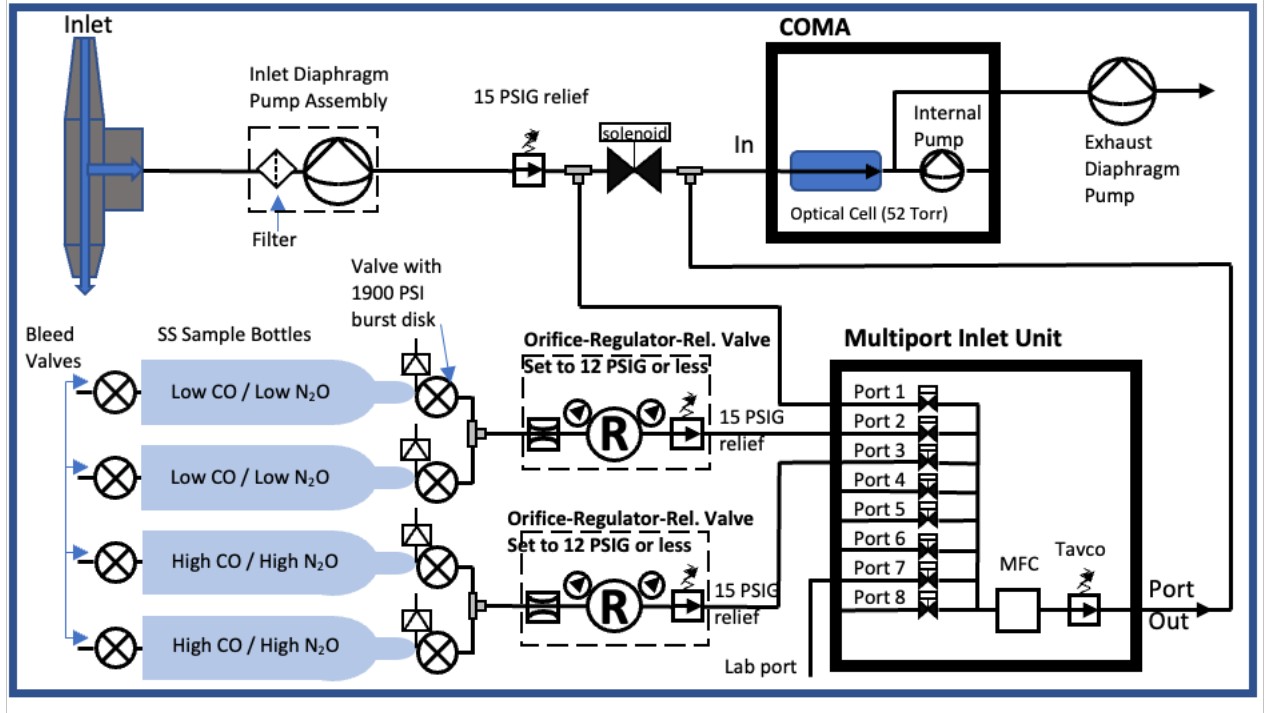

**Figure 3: COMA flow diagram.**

COMA's in-flight calibration system consisted of four Swagelok double-ended stainless steel cylinders (304L-HDF4-1000) with 1800 PSI maximum allowable working pressure (MAWP) and 1000 PSIG maximum operating pressure (MOP). Prior to the ACCLIP campaign deployment, the in-flight cylinders were filled from one of four source cylinders: two primary NOAA Global Monitoring Laboratory whole air standards, certified by the WMO Central Calibration Laboratory for CO (Novelli et a., 1991) and N₂O (Hall et al., 2007) (low CO standard "NOAA 1", (tank # CC745344, CO ~51.77 ppb ±0.94,

N2O ~265.89 ppb ±0.03 and high CO standard "NOAA 2", (tank # CC746190, CO ~164.34 ppb ±1.84, N2O~348.06 ppb ±0.05), and two secondary standards ("Matheson 1", ~200 ppb each of CO and N2O, and "Matheson 2", ~1000 ppb each of CO and N2O). The NOAA standards are referenced on the WMO CO_X2014A scale for CO and the NOAA-2006A scale for N2O. Further details on the calibration scales can be found at https://gml.noaa.gov/ccl/refgas.html. Only two standard cylinders were open at any one time (the other two had valve locks installed on the unopened cylinders to prevent leakage). The two NOAA-filled cylinders were the first to be sampled by COMA during in-flight calibrations before beginning to sample the secondary standards. The standards were deployed during the field campaign, which allowed the in-flight calibration system to be re-filled with the secondary standards and sampled throughout the duration of the ACCLIP deployment.

### 2.1.3 Telemetry

COMA was required to run autonomously onboard the WB-57. The instrument could be controlled (on/off) by the Science Equipment Operator located in the rear cockpit to power cycle the instrument. Bandwidth to communicate with the instrument in-flight was limited. To address this, we developed a communication and processing software for real-time monitoring on the ground, consisting of a Python script which ran on the analyzer to access data files and output 1 Hz user data protocol (UDP) packets. It also included a proconX SERIP-100 serial-to-ethernet converter as backup system, which sent UDP packets for real time monitoring on the ground. In addition, NASA's Mission Tools Suite (MTS) was used to communicate to the ground while in-flight (https://airbornescience.nasa.gov/tracker/).

### 2.2 Instrument Performance

### 2.2.1 Environmental Chamber Testing

Prior to deployment, COMA was extensively tested in the NASA Ames Research Center Engineering Evaluation Laboratory (EEL). The environmental chamber testing plans centered around two main concerns. First, the thermal stability of the instrument's components were tested to ensure safe operation of the laser and other components under UTLS-like conditions. Second, the stability of the instrument was tested by sampling from a standard gas mixture while simulating UTLS conditions.

In the environmental chamber, seven thermocouples were placed inside the COMA instrument, one along the middle dividing panel inside sensor (Figure 4: mid-rib), and at six locations considered critical or at risk of overheating. Figure 4 shows the thermal performance of COMA's individual components during a simulated UTLS flight, when the chamber altitude (pressure) was varied (Figure 4: black line, secondary y-axis). Results from the thermal testing showed that at altitudes up to 18 km, components did not become unstable or overheat outside of the expected operational ranges identified for these components.

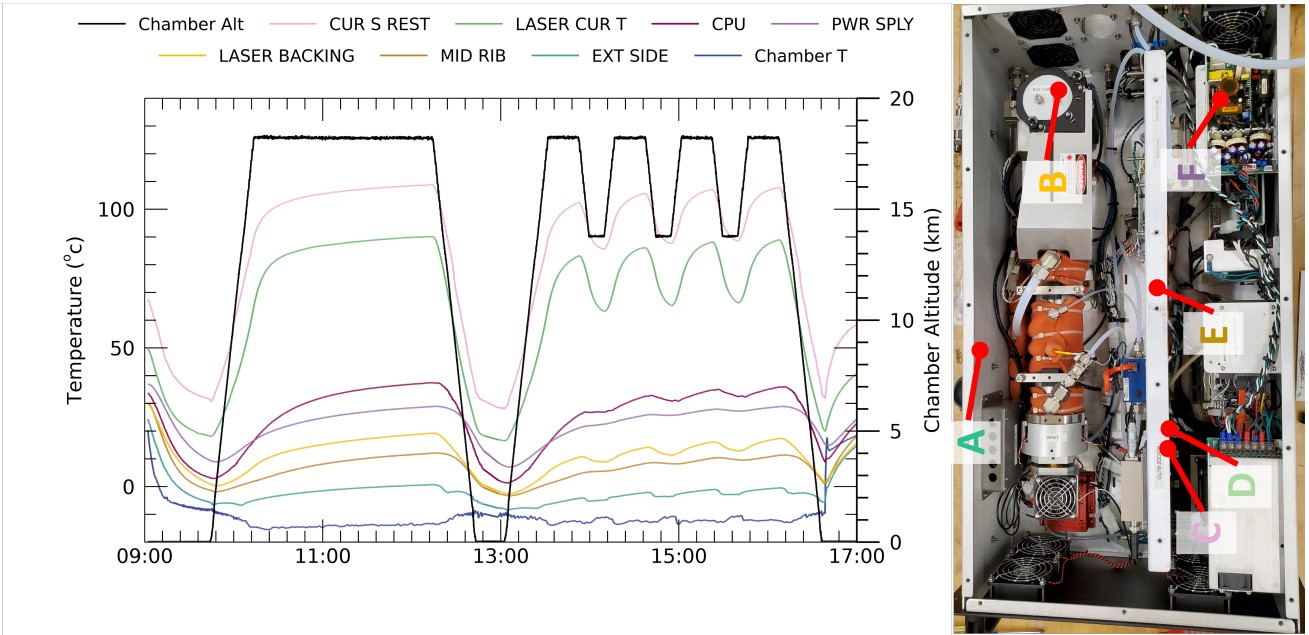

**Figure 4: Environmental chamber timeseries showing COMA's component temperatures during a simulated UTLS flight conducted on 20 May 2022 (right). Corresponding thermocouple locations are shown on the right, external side = EXT SIDE (A), laser backing (B), laser current sensing resistor = CUR S REST (C), laser current temperature = LASER CUR T (D), internal dividing panel = MID-RIB (E), and power supply = PWR SPLY (F).**

Allan deviation (Allan, 1987) was used as a metric of COMA's stability as shown in Figure 5, which presents the Allan deviation as a function of averaging time to illustrate how precision changes based on data averaging. The results show a laboratory 1 Hz precision of ~0.13 ppb for CO and ~0.19 ppb for $N_2O$. Precision improves with increased data averaging during standard sampling in the laboratory (blue) up to ~1000 s, after which additional time averaging has little benefit. In the environmental chamber, under conditions shown in Figure 4, precision is similar at 1 Hz and improves with increased averaging at approximately the same rate of change as in the laboratory up to 10 s, after which the improvements in precision are at a slower rate of change, likely due to noise or additional uncertainty in the measurements due to changing operational (chamber temperature and pressure) conditions.

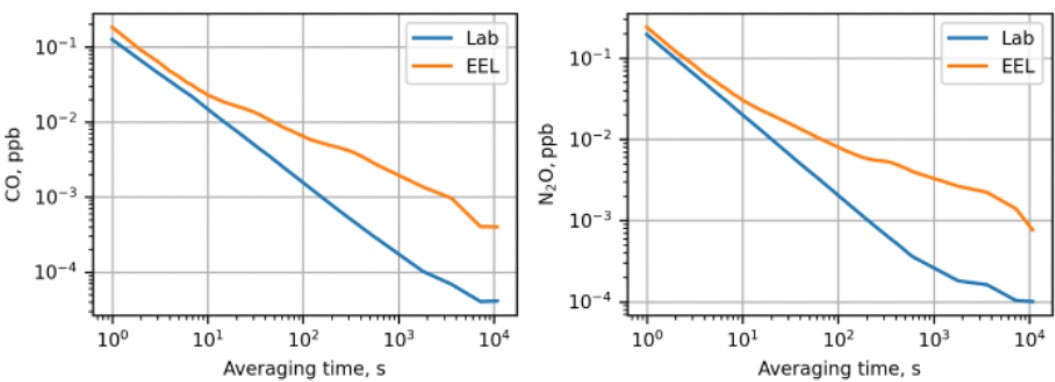

**Figure 5. Allan deviation results, comparing data collected in lab (blue) and an environmental chamber (EEL, orange) during standard sampling for CO (left) and $N_2O$ (right). Under both conditions, 1 Hz precision ~ 0.1 – 0.2 ppb for both gases. See Figure 4 for chamber pressure and temperature details.**

### 2.2.2 Instrument Linearity

Linearity assessments were performed using a flow mixing system equipped with high accuracy mass flow controllers (Alicat Scientific Inc., MC-1SLP-D, (most recent calibration, reported accuracy was ±0.6 % of reading (3 Nov 2021)) and OMEGA Engineering inc., FMA-2602A, (most recent calibration, reported accuracy was ±0.4 % of reading (11 Jul 2021))). A secondary synthetic CO standard (~1000 ppb each of CO and $N_2O$) and a zero-air standard were used to perform the linearity mixing analysis. The linearity assessment for COMA is shown in Figure 6 and demonstrates that COMA is highly linear over a wide range of CO and $N_2O$ mixing ratios. Figure 6 shows COMA to be linear (slope of 1.00) to within 0.1% between 25-1000 ppb CO and linear (slope of 1.00) within 0.1% between 25-850 ppb $N_2O$, with the largest uncertainty equal to the reported accuracies of the flow meters stated above.

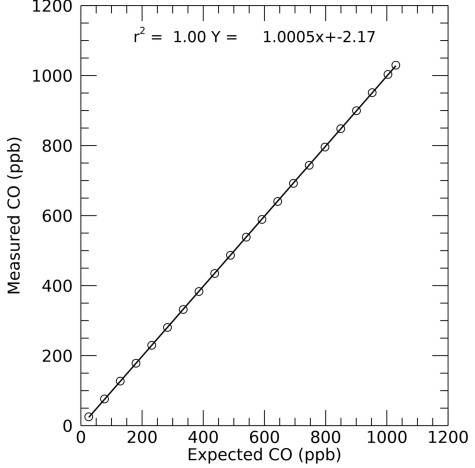
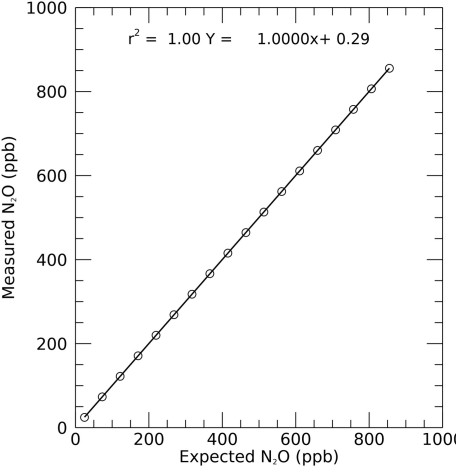

**Figure 6: COMA Instrument linearity for CO (left) and N$_2$O (right) as performed under laboratory conditions on 20 October 2023.**

### 2.2.3 Instrument Calibration and Measurement Uncertainty

Calibration to NOAA standards was applied (see Section 2.1.2) using data collected at ground level throughout the campaign as well as after to help account for time drift in the instrument. Outliers, which deviated from the mean by more than 4 standard deviations were removed. Multivariate linear regressions with time and measured concentration as dependent variables were applied for both CO and N$_2$O. Slight degradation in instrument response was observed over the course of the campaign and was accounted for with the inclusion of an elapsed time term in the final calibrations. The estimation of overall accuracy of CO includes small contributions due to accuracy of the standard gases but was dominated by the residuals remaining after this calibration to NOAA standards. Overall accuracy for CO was determined to be $\pm$ 3.8 ppb over the calibration range. Accuracy for N$_2$O is comprised equally of the contributions from the residuals after calibration to NOAA standards (1.0 ppb) and NOAA scale uncertainty (0.31 %). Given the relatively small range of N$_2$O values observed in the field, overall accuracy in N$_2$O can be approximated by the value of +/- 1.4 ppb, as calculated at N$_2$O = 320 ppb.

The in-flight calibration system ran a cycle of 60 s of low-mixing ratio calibration gas, followed by 60 s of high mixing ratio gas periodically throughout each flight. Figure 7 shows the results from in-flight calibrations for CO (bottom) and N$_2$O (top) during the ACCLIP deployment for primary whole air NOAA standards (left) and secondary synthetic standards (right). Note that only the NOAA standards were used for linear calibration fits; the secondary synthetic standards were only used for internal assessment. The intra- and inter-flight variability among flight calibrations give an indication of the in-flight instrument 1$\sigma$ precision. The standard deviations of observations at three different mixing ratios were seen to vary slightly with mixing ratio as shown in Equation 1 and Equation 2. For example, At 320 ppb N$_2$O precision = 2.3 ppb (equivalent to 0.7 %). At 50 ppb CO precision = 1.4 ppb (equivalent to 2.8 %), while at 200 ppb CO precision = 4.1 ppb (equivalent to 2.1 %). Readers should use discretion if extrapolation of precision is required outside the range used to determine these equations (CO: 48-203 ppb; N2O: 195 – 345 ppb).

$$\mathbf{CO\ precision\ =\ 1.79}x\mathbf{10^{-02} \times CO\ (ppb)\ +\ 0.50} \qquad \text{Equation 1}$$

$$\mathbf{N_2O\ precision\ =\ 8.05}x\mathbf{10^{-03} \times N_2O\ (ppb)\ -\ 0.25} \qquad \text{Equation 2}$$

Overall uncertainty is determined by the square root of the sum of the squares of the accuracy and precision terms. If desired, total uncertainty for each measurement can be calculated from individual terms. Under flight conditions at 320 ppb N$_2$O total uncertainty is 2.7 ppb. At 50 ppb CO total uncertainty is 4.1 ppb; at 200 ppb CO total uncertainty is 5.6 ppb.

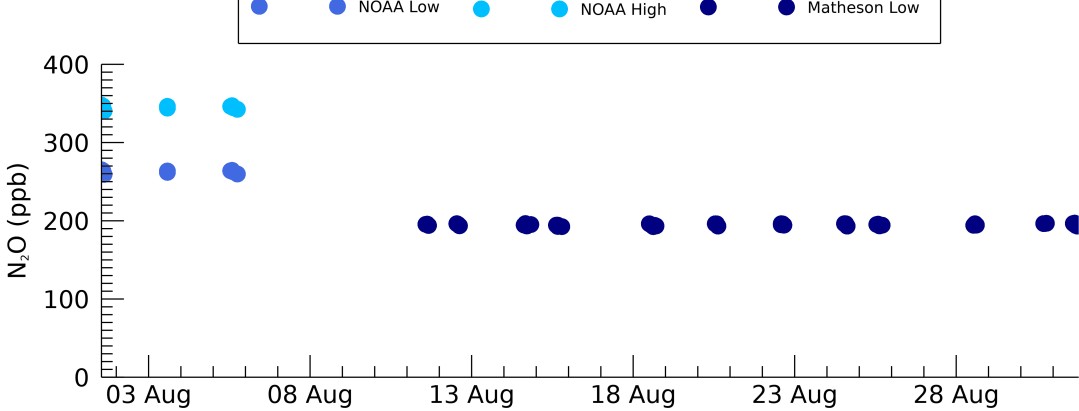

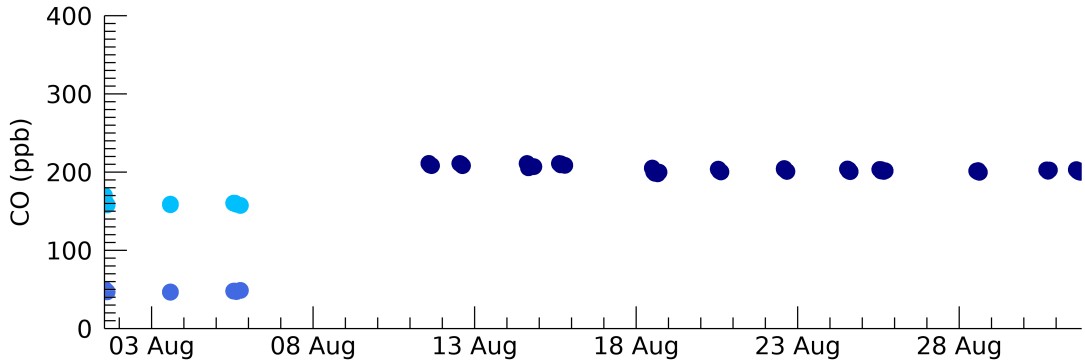

**Figure 7: Mean value from individual in-flight calibration periods for CO (bottom) and $N_2O$ (top) during the ACCLIP deployment for primary whole air NOAA standards (prior to August 8, 2022) and secondary Matheson synthetic standards (after August 11, 2022). The standard deviation of each calibration mean value is significantly smaller than the size of the symbol plotted. Note, the primary whole air standards (NOAA) were sampled during in-flight calibrations prior to August 8, 2022. Once these were exhausted, the secondary standards (Matheson) were used for in-flight calibrations on/after August 11, 2022.**

### 2.2.4 Data Processing

In addition to applying the calibration equations described in Section 2.2.3, several steps were taken to ensure the quality of the reported atmospheric observations of CO and $N_2O$. Measurements recorded during periodic calibration cycles and up to a minute of data before and after were removed, as were measurements before and during take-off and after landing. A time lag is present in the raw data due to combination of physical effects and clock offsets, which were accounted for by determining a median offset for each flight by correlation to an additional CO instrument (COLD2) onboard the WB-57. When COMA's measurement cell pressure deviated from the median pressure by more than 0.25%, $N_2O$ data were omitted, and CO data were omitted on a case-by-case basis. (CO measurements were more robust against cell pressure oscillations, but not immune.) Additional deviations from nominal operating conditions were evaluated on a case-by-case basis.

## 3 COMA In-flight Data

COMA flew onboard NASA's WB-57 on four test flights (TF) from Ellington Field, TX in summer 2021, one functional check flight and two test flights in Ellington Field, TX in July 2022, five outbound transit flights from Ellington Field, TX, USA to Osan Air Base, South Korea, 15 research flights (RF) from Osan Air Base and four return transit flights to Ellington Field, TX, USA. Data is archived and publicly available at the NASA Langley Research Center (LaRC) Distributed Active Archive Center (DAAC) (https://www-air.larc.nasa.gov/cgi-bin/ArcView/acclip.2022).

The WB-57 typically profiled multiple times through the UTLS during each research flight, resulting in vertical profiles of CO and $N_2O$. Figure 7 shows a summary of observations of $N_2O$ and CO reported by COMA plotted by altitude and colored by flight date for ACCLIP campaign research flights from Osan, South Korea. General observations include day-to-day variability in CO and $N_2O$ within the boundary layer (<~2 km). Within the free-troposphere (~2.5 to 12.5 km) $N_2O$ is well-mixed, with little day-to-day variability, however there is more variability observed in CO. The UTLS region (~12.5 to 16 km), shows highly variable CO, with the interception of lofted pollution originating from convective influences over Asia during different flights. In contrast, $N_2O$ remains well-mixed in this altitude region. Decreasing profiles of both CO and $N_2O$ were observed within the stratosphere (>~16 km).

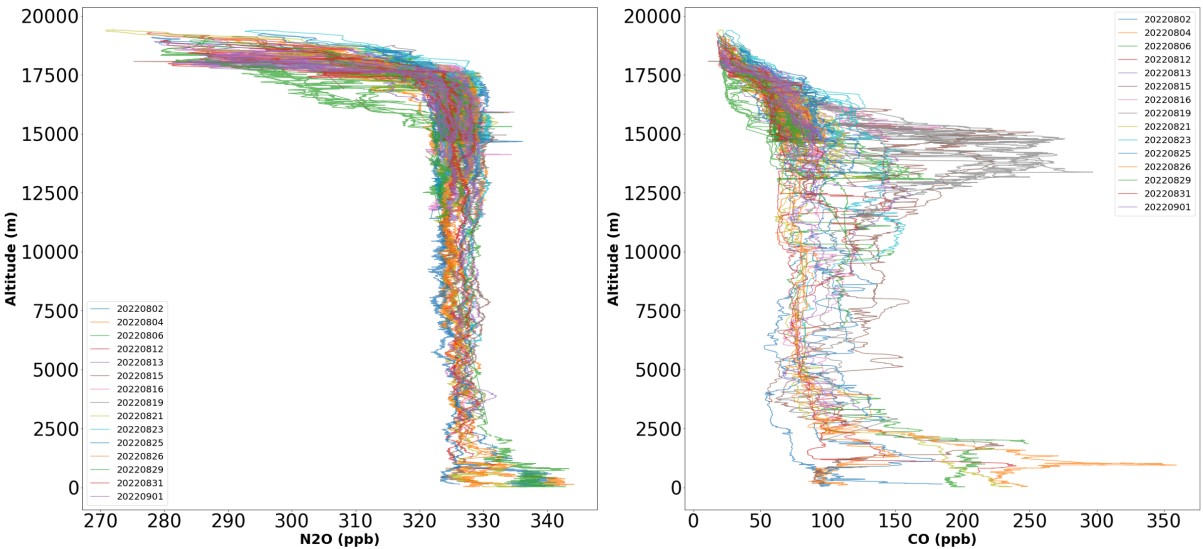

**Figure 8: Vertical profiles of all $N_2O$ (left) and CO (right) measurements taken by COMA during the ACCLIP campaign research flights from Osan, Korea in July and August 2022.**

### 3.1 Data Comparison with COLD2 and ACOS

During the ACCLIP campaign, two other instruments were onboard NASA's WB-57 that measured CO: Carbon Monoxide Laser Detector (COLD) 2 and Airborne Carbonic Oxides and Sulfide Spectrometer (ACOS). COLD2, operated by CNR-INO (CNR National Institute of Optics), is a mid-infrared quantum cascade laser spectrometer that has previously flown on an M55 aircraft during StratoClim (Stratospheric and upper tropospheric processes for better climate predictions)

(Viciani et al., 2018). ACOS is a NOAA-operated, Off-Axis Integrated Cavity Output Spectrometer (ICOS) that measures carbonyl sulfide (OCS) and CO (Gurganus et al., 2024).

Figure 9 shows the CO data comparison as a linear regression and demonstrates an excellent overall agreement between the three CO instruments. The cross plot shows the correlation of the 1 Hz CO measurements with ACOS and COLD2 CO plotted on the vertical axis and COMA CO on the horizontal axis, with a slope of 1.06 for COLD2 ($r^2 = 0.99$) and 1.01 for ACOS ($r^2 = 0.98$), indicating strong agreement throughout the ACCLIP field campaign. Comparisons of the 15 individual flights from the campaign indicate a range in slope of 1.02-1.08 for COLD2 and 0.89–1.10 for ACOS.

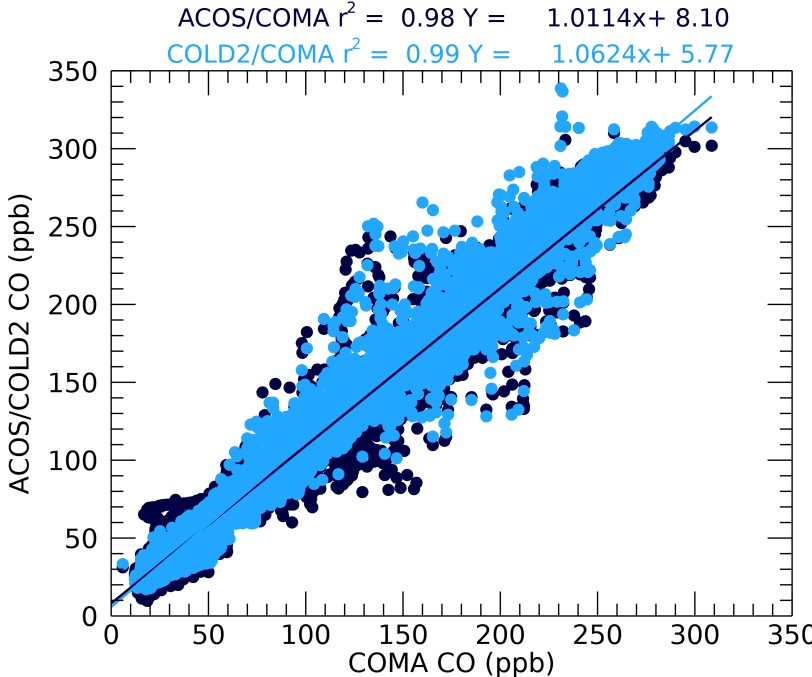

Figure 9: Linear regression of ACOS and COLD2 vs COMA for all 15 ACCLIP science flights.

## 4 Conclusion

The NASA COMA instrument provides high-sensitivity measurements of CO and $N_2O$ from ground-level to an altitude of ~18 km. COMA flew onboard NASA's WB-57 during 24 flights, supporting the ACCLIP field campaign based in Korea during 2022. COMA data from this campaign is archived and publicly available (https://www-air.larc.nasa.gov/cgi-bin/ArcView/acclip.2022).

During ACCLIP, COMA flew primarily between 12 and 19 km within an unpressurized pallet of the WB-57. To operate successfully in these conditions, significant modifications were made to the original laboratory-oriented instrument,

followed by laboratory and environmental chamber testing to simulate instrument behavior under expected flight conditions.

This paper details the COMA design modifications, including the installation into the WB-57 payload pallet required for flight. Details of internal instrument modifications and changes to operational parameters are detailed in Section 2. External modifications to COMA to accommodate flights in the UTLS region included additional heaters and fans to help reduce condensation and for thermal management. An in-flight calibration system was added to the COMA payload to allow an evaluation of COMA in-flight performance, and an MIU to allow gas sampling selection (e.g. inlet, calibration standard, detailed in Section 2.1.2).

COMA instrument performance was assessed through extensive laboratory and environmental chamber testing (detailed in Section 2.2). This testing included assessments of COMA operation under simulated UTLS conditions, linearity assessment, assessment of laboratory and in-flight COMA calibrations, and data comparison of COMA to two independent instruments measuring CO (COLD2 and ACOS). Overall, the instrument achieved an uncertainty during ACCLIP of 2.7 ppb $N_2O$ (at 320 ppb) and 5.6 ppb CO (at 200 ppb). COMA's successful integration aboard NASA's WB-57, demonstrated field performance, and favorable comparison to other independent instruments enable a new CO and $N_2O$ capability for airborne science measurements in the upper troposphere and lower stratosphere.

*Author contributions:* JRP and LTI supervised the project. JE, RRJ, and JRP led design and integration onto the WB-57 aircraft. JBL, IA, JRP, and RRJ contributed towards environmental chamber testing and system design. LMG, JRP, RRJ, RK, LTI, KEO, CD, and ELY were involved in lab and field investigation. KEO, JRP, and LMG curated the COMA data, KEO and LMG wrote software used. LMG and ELY wrote the original draft, which was revised with feedback from all authors.

*Competing interests:* The authors declare that they have no conflict of interest.

**Acknowledgements**

The COMA team was supported by the NASA Earth Science Research and Analysis Program (K. Jucks, J. Kaye), the NASA Postdoctoral Program, and NASA Ames Internal Research and Development funding. Technical contributions from Bob Provencal at ABB, Chris Wilson at NASA Ames and staff at the Ames Engineering Evaluation Laboratory (EEL) are greatly appreciated. We gratefully recognize the efforts of all involved in the ACCLIP mission, from inception to project management to aircraft support and beyond. This work benefits from COLD2 data provided by Francesco D'Amato, Silvia Viciani and Giovanni Bianchini at CNR-INO. The COLD2 team was supported by ESA, within the frame of Project SVANTE-QA4EO. The ACOS data provided by Colin Gurganus at NOAA CSL was supported by the NOAA Chemical

Sciences Laboratory and the University of Colorado through the Cooperative Institute for Earth System Research and Data
Science (CIESRDS).

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
