# Peer review of "Description and Validation of a Carbon Monoxide and Nitrous Oxide Instrument for High-Altitude Airborne Science (COMA)"

_EGUsphere, 2025_

## Referee Comment (RC2)

This article is about the customization and operation of the Carbon monOxide Measurement from Ames (COMA) instrument onboard NASA's high-altitude WB-57 research aircraft. The paper is well within the scope of AMT. In a good wording, the authors describe a novel technology created to further provide important stratospheric measurements of CO and N2O for altitudes above ~ 12 km that can not be reached by more traditional research aircraft used in field campaigns or commercial aircraft. This technology was deployed in operation during the ACCLIP experiment and presents a unique opportunity to bring to the science community original results on the pollution transport processes within the Asian summer monsoon. I recommend publication with minor revisions to address the questions below :

Line 68 : Please, confirm that COMA is based on the ABB ltd GLA251 Series instrument. I can not find any reference on their commercial website. Please, provide a reference for the original instrument specifications.

Figure 1 : Could you make it bigger? Please, specify the units for the length 17.81 and 12. Also, avoid shortened words if they are not described earlier e.g "cal gas", "Pallet Cross Sect.", "Structure to CL Dist.", etc … Does "regulators (2)", Sample Gas pumps (2)" means that there are 2 regulators (of what) and 2 pumps? In "Clearance below Pallet #4 (6 in.)", for what stands #4? Maybe, you should add more details in the description paragraph below the Figure 1 to better understand what we see.

Section 2.1.2 Flow system : I find the paragraph too minimalist and seams to me incomplete. Please, describe the need of the exhaust diaphragm pump and the internal pump, where goes the air after sampling? What is the required flow for the measurement cell? Do you monitor it?

Figure 4 : It is difficult to see where the arrows point on the photo. Can you make the photo bigger?

Figure 9 : You should plot the ratio or the relative difference of concentrations rather than the absolute concentration time-series. Do you get consistent results for the other flight missions? If not, what could explained it? Were they connected to separate intake inlet?

Conclusion : Please, add more results details. Summary the technical challenges that were solve to successfully operates the COMA instrument up to 18 km.

Line 247 : In the abstract, you wrote 5.9 ppb at (200 ppb) ...

---

## Author Response (AR1)

**Response to referee #1**
This article is about the customization and operation of the Carbon monOxide Measurement from Ames (COMA) instrument onboard NASA's high-altitude WB-57 research aircraft. The paper is well within the scope of AMT. In a good wording, the authors describe a novel technology created to further provide important stratospheric measurements of CO and N2O for altitudes above ~ 12 km that cannot be reached by more traditional research aircraft used in field campaigns or commercial aircraft. This technology was deployed in operation during the ACCLIP experiment and presents a unique opportunity to bring to the science community original results on the pollution transport processes within the Asian summer monsoon. I recommend publication with minor revisions to address the questions below:
*Response:*

Line 68: Please, confirm that COMA is based on the ABB ltd GLA251 Series instrument. I cannot find any reference on their commercial website. Please, provide a reference for the original instrument specifications.
*Response: You are right this is very hard to find on the ABB website. I checked the operating manual we have for the original instrument and confirm it is 'GLA251-NO2CM' is the model number we were given for the original N2O/CO analyzer. I assume this model number has been retired/updated as the current model number for a similar instrument is GLA351-N2OCM which can be found on the ABB website.*

Figure 1: Could you make it bigger? Please, specify the units for the length 17.81 and 12. Also, avoid shortened words if they are not described earlier e.g "cal gas", "Pallet Cross Sect.", "Structure to CL Dist.", etc ... Does "regulators (2)", Sample Gas pumps (2)" means that there are 2 regulators (of what) and 2 pumps? In "Clearance below Pallet #4 (6 in.)", for what stands #4? Maybe, you should add more details in the description paragraph below the Figure 1 to better understand what we see.
*Response: We have added more context to the legend of Figure 1, adding additional details on the layout and dimensions of COMA within the WB-57 payload bay. We have also increased the size of Figure 1.*

Section 2.1.2 Flow system: I find the paragraph too minimalist and seems to me incomplete. Please, describe the need of the exhaust diaphragm pump and the internal pump, where goes the air after sampling? What is the required flow for the measurement cell? Do you monitor it?
*Response: We have added more details to this section. With regard to flow rate -* Flow rate is not measured/recorded by COMA, other measurements which are indicative of operation/flow are measured including sample cell pressure which is used as a primary indicator of instrument operation (i.e. some deviations in cell pressure were observed on some descents as the instrument was cold-soaked and if the instrument descended into particularly humid conditions this would causing icing within the lines, which would block flow, impacting the cell pressure and as we used this variable in our post flight analysis, this data would be flagged and subsequently removed from the final dataset).

Figure 4: It is difficult to see where the arrows point on the photo. Can you make the photo bigger?
*Response: We have increased the line thickness on the arrows to make this easier to identify.*

Figure 9: You should plot the ratio or the relative difference of concentrations rather than the absolute concentration time-series. Do you get consistent results for the other flight missions? If not, what could explained it? Were they connected to separate intake inlet?
*Response: We have re-plotted Figure 9 to include a comparison of COMA with COLD2 and ACOS instruments during the entire ACCLIP flight data, which shows a more thorough comparison of the different instruments. By doing this we have had to remove the timeseries plot as there is not a constructive way of showing this when the entire dataset is used. We feel the ratio plot is the best method to display the intercomparison of ACCLIP flight data from the three independent instruments.*

Conclusion: Please, add more results details. Summary the technical challenges that were solve to successfully operates the COMA instrument up to 18 km.
*Response: We have updated this section to include a more detailed summary of this study.*

Line 247 : In the abstract, you wrote 5.9 ppb at (200 ppb) ...
*Response: 5.6 is correct, we updated this in the abstract. Thank you for spotting this typo.*

**Response to Referee #2**

This manuscript describes the NASA Ames COMA instrument, a high-altitude airborne sensor for measurement of carbon monoxide and nitrous oxide gas concentrations. The manuscript describes the customization and refining of the core commercial sensor and laboratory and chamber testing. Data from the ACCLIP science campaign is discussed, along with carbon monoxide intercomparisons with two other sensors during that campaign. The paper is well within the scope of AMT, and presents new, novel measurement technology. The paper is clearly written with some small exceptions. I recommend publication after addressing some minor changes detailed below:

*The authors would like to thank the referee for their reviews and comments; we have responded to each comment separately below; our response is in italics.*

Line 93: Last sentence seems unnecessary since this describes the next section.
*Response: This has been deleted.*

Figures 2 & 4: The choice of colors may be challenging for color-blind individuals. I would recommend altering the colors or adding dashed/dotted lines. At a minimum, reorganizing the legend in the same order as the color traces vertically would help.

*Response: Figure 2 and 4 have been updated with an update on colors, legend layout/ordering and legend description to address these concerns.*

Line 120: I would recommend citing the calibration source papers directly rather than the website.
*Response: the website gives the most update information on the scales https://gml.noaa.gov/ccl/refgas.html . We have added additional references for NO2 scale from 2007 (doi:10.1029/2006JD007954.) and CO from (https://agupubs.onlinelibrary.wiley.com/doi/abs/10.1029/91JD01108).*

Line 153: Which segment in figure 4 was used to perform the Allan variance calculation? Was it the entire timeseries? Seems like this would be somewhat of a worst case scenario, since most UTLS missions would have a single ascent to altitude with some profiling up high (similar to the latter half of the chamber timeseries in Fig 4). A little more information would be useful for context.
*Response: The Allen variation calculation does indeed reflect worst case scenario and was based on the timeseries in Figure 4.*

Sect. 2.2.2 & Fig 6: this section is a bit light and imprecise. Linearity is always with caveats with respect to uncertainty. How accurate are the flow controllers? Are they new with factory traceable calibrations or were they recalibrated for the experiment? I suppose the uncertainty in the standard would cancel out when just proving linearity, but the mixing errors are definitely important. Usually one can say something like "instrument is linear to within X% between MM-NN ppm".
*Response: We have added more context to this section to better describe the mixing system and its traceability.*

Line 174: "Slight degradation…was accounted for." How was it accounted?
*Response: We have expanded on this sentence to include: "Slight degradation in instrument response was observed over the course of the campaign and was accounted for with the inclusion of an elapsed time term in the final calibrations."*

Line 175: maybe change "small terms due to accuracy of the standard gases" to "small contributions due to the accuracy of the standard gases", it took me awhile to figure out what a small term was referencing
*Response: Done.*

Line 178: I think there is a word missing here…maybe "equally between the residuals"?
*Response: We have changed this sentence to read 'Accuracy for $N_2O$ is comprised equally of contributions from the residuals after calibration to NOAA standards'*

Line 189: Is there any theory as to why the precision varies
*Response: Instrument precision is impacted by both internal (ability to maintain sample cell pressure, flow rate, internal temperatures etc) and external variables (temperature, humidity,*

*variation etc). We ran laboratory and chamber tests and an in-flight calibration system to be able to define COMA's overall uncertainty to the best of our ability.*

Eq. 1&2: this might be more readable if the slopes were expressed as percents? That's what I'm typically looking for here…just a suggestion.
*Response: We have left these equations as is as we don't think that adding one more operation to get to percents will make it easier to understand.*

Figure 7: what happened after the Aug 8th so that there are no longer 2 point NOAA gas calibrations?
*Response: We filled the in-flight calibration system with the NOAA gases (primary standards) in our laboratory, prior to field deployment in Korea. We had no means of re-filling the NOAA standards once we left Ellington Air Force Base in Texas, but did have the secondary (Matheson) standards, which had been shipped ahead to the field site. We first used just NOAA standards to run the in-flight calibration cycles. Once we had exhausted these, we switched to the Matheson standards. We have added as a note to Figure 7 caption.*

Line 214: Cite data DOI?
*Response: This is cited in the following section – 3. COMA In-flight Data.*

Sect 3.1/Figure 9: Why only data from one flight? I think it is important to include all data from the campaign unless there are flights where this is not possible (e.g. missing data), along with a discussion of where they disagree and what that might mean. I also usually like either ratio or difference plots for intercomparisons rather than full scale concentration timeseries, as it highlights differences more.
*Response: We have re-plotted Figure 9 to include a comparison of COMA with COLD2 and ACOS instruments during the entire ACCLIP flight data, which shows a more thorough comparison of the different instruments. By doing this we have had to remove the timeseries plot as there is not a constructive was a showing this when the entire dataset is used. We feel the ratio plot is the best method to display the intercomparison of ACCLIP flight data from the three independent instruments.*

Conclusion: the summary is a little slim, I would add more summa  on about the laboratory experiments and calibration.
*Response: We have re-written and added to this section to provide a more detailed summary of this study.*

**Response to Referee #3**

Yates et al. present an instrument to measure CO and $N_2O$ aboard high altitude aircraft. They describe the modifications they made to a commercial analyzer in order for it to perform on an aircraft platform that samples up to 18 km in the atmosphere and in very cold and warm/humid conditions. They compare their CO measurements from one campaign on a NASA WB-57 platform to two other instruments that measured CO.  Correlation slopes between the other

two instruments vs. the instrument described in depth in this manuscript had slopes that ranged from 1.10–1.15 and 0.94–1.10, respectively, depending on the flight that was compared.

Overall, this paper described modifications to a commercial instrument that could be helpful to the scientific community aboard high-altitude aircraft.  However, I think the authors should go into more details on both the modifications and for the other CO instruments that were compared to.  Therefore, I think major revisions are necessary.

*Response: The authors would like to thank the referee for their review and comments; we feel the revised manuscript has been significantly improved as a result. We have responded to each comment separately below, our response is in italics.*

First, more detail is needed for the inlet diaphragm pump assembly.  For example, the I was surprised that the calibration bottles were not inserted into the sample flow upstream of the diaphragm pump.  Were any tests done in the lab to show that this pump did not affect the CO or $N_2O$ measurement?  What material were the diaphragm parts and seals made of?

*Response: We have added more details to the 'flow system' section. Re calibration bottle's location: The inlet diaphragm pump was installed upstream of COMA to increase the inlet pressure when sampling ambient air at high altitudes. Multiple testing in the environmental test chamber allowed us to evaluate the instruments performance under reduced pressure (and temperature). We found an ideal operating cell pressure of ~52 Torr. The upstream pump allows for this cell pressure to be maintained, up to 18 km altitude with existing instrument orifices/ vacuum controller conductance's. Because outlet pressure from the calibration bottles is maintained it was not required to flow through the inlet diaphragm pump) prior to sampling. Re testing/pump: No prior testing was done on the pumps effects on CO/N2O. However, upstream KNF's pump have been used in prior studies measuring CO and N2O (e..g. [https://amt.copernicus.org/articles/12/637/2019/amt-12-637-2019.pdf](https://amt.copernicus.org/articles/12/637/2019/amt-12-637-2019.pdf))*

Along those lines, I am not aware of people using Teflon FEP tubing to measure CO and $N_2O$. Are there any lab tests the authors can point to that show that this material does not affect CO and $N_2O$?

*Response: We did not do any lab tests on this. We were careful to keep all tubing to a minimum and used Teflon FEP tubing in locations where we needed moveability/flexibility (i.e. connecting/disconnecting to inlet). The same tubing of similar lengths is used in the calibration line and inlet/air sample line. So if it is true that there are some effects they would be in both lines and any offsets would be accounted for by the calibrations.*

I also had some questions about the flow diagram in Figure 3.  What is the purpose of Port 1?  Is it the default flow path?

*Response: Port 1 in this diagram is the default sample line.*

line 68-73, how was cell pressure maintained?  Were the authors using the slow flow path that originally comes with an LGR/ABB instrument?  What was the flow rate in flight?

*Response: Cell pressure is maintained by flow though the inlet diaphragm pump, along with an internal (to COMA) pump and the external (to COMA) exhaust pump (both of which are provided by the manufacturer). COMA contains internal valves that maintain a pressure of ~52 Torr within the sampling cell. We have added additional text to the 'flow system' section to better describe this.*

*Flow rate is not measured by COMA and therefore not recorded in-flight, other measurements which are indicative of operation/flow are measured including sample cell pressure which is used as a primary indicator of instrument operation (i.e. some deviations in cell pressure were observed on some descents as the instrument was cold-soaked and if the instrument descended into particularly humid conditions this would causing icing within the lines, which would block flow, impacting the cell pressure and as we used this variable in our post flight analysis, this data would be flagged and subsequently removed from the final dataset).*

Figure 2, how are the "Solenoid" and "Ext front" temperatures below the ambient temperature?  Also, the text suggests this is in flight data, but the figure caption says the black trace is the environmental chamber pressure altitude?

*Response: We have re-worded the misleading caption; the altitude is indeed flight altitude and data is from flight data on 21 Aug 2022. We have re-plotted the data and noticed a mis-labeled legend, Ambient temperature has been re-plotted and is now very much lower than the other temperatures.*

line 173, I'm not sure why calibrations that differed by more than 4 standard deviations were removed?  Wouldn't this reflect instrument performance and need to be retained?

*Response: Instrument precision is impacted by both internal (ability to maintain sample cell pressure, flow rate, internal temperatures etc) and external variables (temperature, humidity, variation etc). We ran laboratory and chamber tests and an in-flight calibration system to be able to define COMA's overall uncertainty to the best of our ability and to define a data filter*

*that is applied to all data (not just calibration data) to remove spurious spike/deviations in N2O and CO that are not reflective of the sample. This filter was the best performing filter to identify (and remove) data spikes while retaining data representative of the sample. We added some text to section 2.2.3 to explain this.*

Figure 7, why the switch from NOAA standards to Matheson standards?  Could the authors discuss this switch?

*Response: We filled the in-flight calibration system with NOAA (primary) standards in our laboratory, prior to field deployment in Korea. The secondary (Matheson) standards were deployed with the field campaign. The two NOAA-filled cylinders (primary standards) were the first to be sampled by COMA during in-flight calibrations before moving on to sample the secondary standards. The deployed secondary standards were used to re-fill the in-flight calibration system to ensure there were calibrations throughout the campaign (the primary standards were not deployed). We have added some additional context to the paper (Section 2.1.2).*

Figure 9, could the authors use a different color for COMA?  It is hard to distinguish the dark blue from the light blue and/or black trace.

*Response: We have re-plotted Figure 9 to include a comparison of COMA with COLD2 and ACOS instruments during the entire ACCLIP flight data, which shows a more thorough comparison of the different instruments. By doing this we have had to remove the timeseries plot as there is not a constructive was a showing this when the entire dataset is used (which was the plot referred to in this comment). We feel the ratio plot is the best method to display the intercomparison of ACCLIP flight data from the three independent instruments.*

line 229, could the authors add a short description of which groups operate COLD 2 and ACOS?  And how were they calibrated?  Did they get their standards from NOAA?

*Response: We have added the group names to the text in Section 3.1. We also provide references to publicly available papers that further describe the instruments.*

line 236, since the authors use two decimal places for the overall comparison, it is probably best to use two to describe the August 29 comparison

*Response: We have updated this text and Figure 9 to provide comparisons of the instrument performance over the entirety of the ACCLIP campaign. Decimal places have been updated also.*

Typos/Grammar suggestions

line 115, NOAA ESRL no longer exists, and it is misspelled "ERSL". I would use "NOAA's Global Monitoring Laboratory".

*Response: Updated.*

line 121, the URL is missing "a.". Should be …gml.noaa.gov…

*Response: Updated*

lines 93 and 129, don't need to hyphenate "in-flight"

*Response: We have kept this hyphenated for consistency throughout the paper.*

line 178, perhaps say "equally comprised of the residuals…"

*Response: This section has been updated.*

line 217, remove comma after "$N_2O$"

*Response: Updated*

---

## Author Response (AR2)

**Response to Editor report #1**

Thank you for your comments, we have responded to them in *italics* below.

1. Related to Reviewer #1, Sect. 2.2.2/Fig. 6 comments: I'm not sure you can know the linearity better than the flow controllers. It seems like the slopes should at least have a 0.4% uncertainty based on the best flow controller.

*Response: We have added a statement to the relevant text in lines 179-181 to address this: The linearity assessment for COMA is shown in Figure 6 and demonstrates that COMA is highly linear over a wide range of CO and $N_2O$ mixing ratios. Figure 6 shows COMA to be linear (slope of 1.00) between 25-1000 ppb CO and linear (slope of 1.00) between 25-850 ppb $N_2O$, with the largest uncertainty equal to the reported accuracies of the flow meters stated above.*

2. somewhere qualify that Equation 2 is good for N2O > x, since the precision will eventually become 0 and then negative at low N2O concentrations?

*Response: We have added a statement to the relevant text in lines 206-208 to address this: At 50 ppb CO precision = 1.4 ppb (equivalent to 2.8 %), while at 200 ppb CO precision = 4.1 ppb (equivalent to 2.1 %). Readers should use discretion if extrapolation of precision is required outside the range used to determine these equations (CO: 48-203 ppb; N2O: 195 – 345 ppb).*

3. I'm curious how the full comparison of COMA with COLD2 with a slope of 1.06 is lower than any individual flight (1.10–1.15). By deleting the ranges, the authors are deleting possible relevant information related to how the COLD2 instrument performed. Please add the ranges back into the text.

*Response: We have added the ranges back, in addition to the overall, average. Note the values have been updated to reflect comparisons run using the most recent and finalized COMA data product, obtained here: https://www-air.larc.nasa.gov/cgi-bin/ArcView/acclip.2022#PODOLSKE.JAMES/*